# Yttrium Oxyfluoride Coatings Deposited by Suspension Plasma Spraying Using Coaxial Feeding

**Jaehoo Lee, Seungjun Lee, Heung Nam Han, Woongsik Kim \* and Nong-Moon Hwang \***

Department of Materials Science and Engineering and Research Institute of Advanced Materials, Seoul National University, Seoul 08826, Korea; caspiel5202@snu.ac.kr (J.L.); lsj2914@snu.ac.kr (S.L.); hnhan@snu.ac.kr (H.N.H.)

\* Correspondence: woongs1@snu.ac.kr (W.K.); nmhwang@snu.ac.kr (N.-M.H.)

**Abstract:** The recently discovered yttrium oxyfluoride (YOF) coating has been found to be a highly promising plasma-resistant material which can be coated onto the inner wall of the dry etching chambers used in the manufacturing of the three-dimensional stacking circuits of semiconductors, such as vertical NAND flash memory. Here, the coating behavior of the YOF coating which was deposited by suspension plasma spraying was investigated using a high-output coaxial feeding method. Both the deposition rate and density of YOF coatings increased with the plasma power, which was determined by the gas ratio of $Ar/H_2/N_2$ and the arc current. The coating thicknesses were $58 \pm 3.4$, $25.8 \pm 2.1$, $5.6 \pm 0.6$, and $0.93 \pm 0.4$ μm at plasma powers of 112, 83, 67, and 59 kW, respectively, for 20 scans with a feeding rate of the suspension at 0.045 standard liters per minute (slm). The porosities were $0.15\% \pm 0.01\%$, $0.25\% \pm 0.01\%$, and $5.50\% \pm 0.40\%$ at corresponding plasma powers of 112, 83, and 67 kW. High-resolution X-ray diffraction (HRXRD) shows that the major and minor peaks of the coatings which were deposited at 112 kW stem from trigonal YOF and cubic $Y_2O_3$, respectively. Increasing the flow rate of the atomizing gas from 15 slm to 30 slm decreased the porosity of the YOF coating from $0.22\% \pm 0.03\%$ to $0.07\% \pm 0.03\%$. The Vickers hardness of the YOF coating containing some $Y_2O_3$ deposited at 112 kW was $550 \pm 70$ HV.

**Keywords:** yttrium oxyfluoride (YOF); plasma-resistant material; suspension plasma spraying (SPS); dense YOF coating; plasma power; atomizing gas

## 1. Introduction

In order to manufacture a semiconductor circuit, etching, cleaning, and deposition processes are repeated. During these processes, the inside of the processing chamber is exposed to corrosive plasma [1–4]. A highly reactive gas such as $CHF_3$ is used during the dry etching process, and a fluorine-based gas such as $NF_3$ is used in the cleaning process to remove the reaction products generated from the inside walls of the equipment [5]. During this process, the coating material on the inner wall of the equipment is eroded and etched by the plasma gas. As a result, particles are generated [6]. These unwanted particles act as contaminants. As the semiconductor industry advances and the incorporation of integrated circuits (ICs) on wafers reaches its limit, semiconductor ICs are more commonly manufactured by stacking circuits in three dimensions, as in three-dimensional vertical NAND (3D V NAND) technology [7]. As the number of stacks is increased, the wafer becomes increasingly more exposed to harsh plasma environments, and the particle generation problem becomes more serious. Therefore, the coating of the inner wall of the semiconductor equipment with a plasma-resistant material has been attempted to minimize the etching of the coating and particle contaminant generation.

In the past, silicon-based materials, which have high hardness, high dielectric strength levels, high wear resistance, and good chemical stability, were used as a plasma-resistant material. However, these materials react with fluorine in the plasma, leading to the production of contaminants

[8–10]. Therefore, silicon-based materials were replaced by the more plasma-erosion resistant $Al_2O_3$ [11–13].

$Al_2O_3$ was again replaced with $Y_2O_3$, which is chemically more stable and it has a lower enthalpy of formation than $Al_2O_3$ [5]. However, when the $Y_2O_3$ coating layer is exposed to fluorine plasma, fluorine-containing particles are generated in the gas phase and land on the surface, contaminating the wafer surface [14]. For this reason, $YF_3$ was suggested as a new material to replace $Al_2O_3$ and $Y_2O_3$ because it is more resistant to plasma and has higher dielectric strength than $Y_2O_3$ [14,15].

Recently, it was discovered that, when the $Y_2O_3$ coating layer reacts with fluorine plasma, an yttrium oxyfluoride (YOF) layer is formed on the surface, with this surface being highly plasma-resistant because it contains metal and fluorine components which are chemically stable. The enthalpy of formation of the metal-oxygen bond of YOF, the oxidized form of $YF_3$, which is $-392$ kJ·mol$^{-1}$, is smaller than that of $Y_2O_3$, which is $-318$ kJ·mol$^{-1}$. Therefore, YOF is more chemically stable than $Y_2O_3$. Moreover, the YOF layer can effectively inhibit particle generation [4,5].

It was also reported that YOF has a denser crystalline structure and thereby higher hardness and better corrosion resistance than $YF_3$ [16]. Therefore, as a substitute for $Y_2O_3$ and $YF_3$, YOF materials have attracted much attention. There are typically two ways to deposit YOF coatings: atmospheric plasma spraying (APS) and suspension plasma spraying (SPS). APS can produce a thick coating with high coating efficiency, but the coating tends to have numerous cracks and pores, making it vulnerable to plasma erosion [17]. SPS can overcome this disadvantage of APS, producing a dense coating [18–20]. Because the size of the feedstock particles in the droplet is as small as 10 μm or less, the splat size of the coating layer can be assuredly small [21]. Because coating YOF with the SPS method is currently in its early stages, the properties of the YOF coating are scarcely known. To the best of our knowledge, YOF coatings have not yet been implemented in the manufacturing process of semiconductors, although tests are underway regarding their adoption.

In spite of the advantage of SPS, it has a critical problem of consuming lots of energy in evaporating the solution. Because of this energy consumption, the power should be high enough to melt the particles that are suspended in the solution. However, the plasma power cannot be increased too high because of the gouging of the electrode. To overcome this difficulty, we used the Axial III torch (Mettech), which has three cathodes and anodes with an axial feeding system. In this respect, the axial feeding has a much more advantage than the radial feeding commonly used in the SPS process due to the effective penetration of suspension into the core zone of the plasma jet. Besides, the high plasma power of the Axial III multi-electrodes has beneficial effects on particle melting and coating behavior.

## 2. Experimental Procedure

### 2.1. Suspension Plasma Spray (SPS) System

Suspension plasma spray equipment (Axial III, Mettech, Surrey, BC, Canada) is used to coat YOF coatings on two types of substrates. As shown in Figure 1, this equipment has the advantage of supplying the suspension to the plasma jet along the central axis of the plasma gun by axial feeding using a co-axial injector. When considering the penetration of the suspension into the plasma jet and the heat transfer between the suspension and the plasma jet, axial feeding is more efficient than radial feeding, in which the suspension is supplied perpendicular to the plasma jet. Meillot et al. [22] reported that the coating efficiency is related to the suspension penetration into the plasma jet. As they used radial feeding, the coating efficiency was sensitive to the suspension injection pressure. However, with axial feeding, the axially supplied suspension can penetrate the core zone of the plasma jet, which makes the coating efficiency higher than that of radial feeding.

This equipment has three cathodes and anodes which are operated by three independent power supplies, which makes it possible to generate a high power without the cathodes being eroded by the plasma [23]. In this sense, this equipment has an advantage over the conventional plasma spray equipment, which typically has a single cathode and anode. Another advantage of this equipment is that the three plasma torch voltages fluctuate independently, which makes plasma gas velocity

variations less sensitive to voltage fluctuations [22]. Typically, an Ar–$N_2$–$H_2$ or Ar–$N_2$–He gas mixture is used with possible electric power up to 120 kW. Argon was supplied in the direction perpendicular to the feeding line as an atomizing gas at a flow rate of 15 or 30 standard liters per minute (slm) in order to break up the droplets.

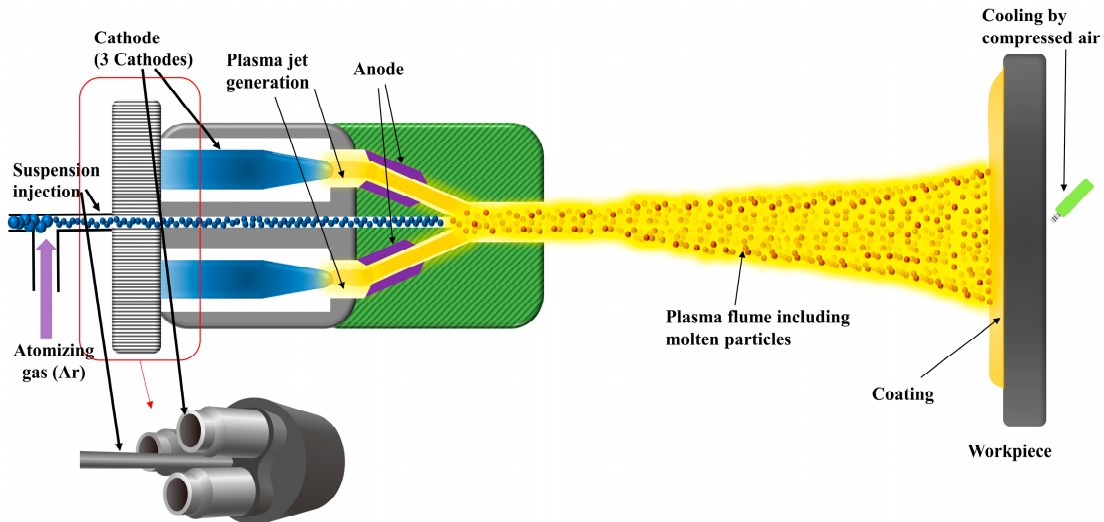

**Figure 1.** Schematic of the Axial III SPS system [24].

## 2.2. Feedstock Materials and Specimens

A commercial $Y_5O_4F_7$ suspension (Nippon Yttrium Co., Ltd., Omuta, Fukuoka, Japan) was used in the SPS coating process. The solvent is deionized water and the average size of the $Y_5O_4F_7$ particles is 3 µm with a suspension solid concentration of 10 wt.%. The plasma power and flow rate of the atomizing gas were varied to study their effects on the coating behavior of the YOF coatings. YOF coatings were deposited onto Al 6061 substrates of two different sizes of 10 mm³ × 10 mm³ × 10 mm³ and 50 mm³ × 50 mm³ × 10 mm³. To improve the adhesion strength of the coatings, the surface of the Al substrate in each case was sandblasted to have a surface roughness average (Ra) value in the range of 2.3–2.8 µm by alumina particles (white fused alumina # 100, Dae Han Ceramics Co., Ltd., Yeongam-gun, Jeollanam-do, Korea) less than 254 µm in size. The Ra of the Al substrate was measured by surface roughness tester (MITUTOYO SJ-210, Kawasaki-shi, Kanagawa, Japan). Furthermore, the substrate was preheated with the plasma flame by scanning the entire surface of the substrate twice before the suspension was loaded onto the plasma jet. The temperature of the substrate was measured by a pyrometer (Fluke, 568 IR thermometer) and it was found to be 107 °C immediately after the substrate was preheated, and the temperature of the substrate was measured to be 277 °C immediately after the coating. The opposite side of the substrate was cooled by air at a distance of ~100 cm in order to prevent the substrate damage due to the high-temperature plasma. We were guided by the reference [6] to choose the distance, which used the same Axial III plasma torch as our experiment. The air flux coming from the air gun was wide enough to cool the 10 mm³ × 10 mm³ × 10 mm³ and 50 mm³ × 50 mm³ × 10 mm³ substrates. The transverse speed of the plasma spraying gun was 1000 mm/s, and 20 coating cycles were used. For the 10 mm × 10 mm × 10 mm substrate, the horizontal and vertical spans of the gun trajectory were 460 and 40 mm, respectively. For the 50 mm × 50 mm × 10 mm substrate, the horizontal and vertical spans of the gun trajectory were 460 and 90 mm, respectively. The vertical increment of the gun position for every horizontal movement was 3 mm. The stand-off distance, referring to the distance between the gun and the substrate, was 50 mm. These conditions were chosen based on the literature [18,25].

*2.3. Analysis Methods*

For microscopic analyses: the coated surface in each case was cleaned with ethanol and the cross-section of the YOF coatings was fixed by hot mounting and polished to 1 μm. A field-emission scanning electron microscope (FE-SEM, SU-70, Hitachi, Tokyo, Japan) was used to analyze the microstructure and morphology of the YOF coatings. The crystal structure of the YOF coatings was analyzed while using a high-resolution X-ray diffractometry (HRXRD, SmartLab, Rigaku, The Woodlands, Texas, USA). An image analysis program (ImageJ developed by the National Institutes of Health and the laboratory for Optical and Computational Instrumentation) was used to analyze the porosity of the cross-sections of the coatings [26].

In each case, the hardness of the coating was measured by a Vickers hardness tester (Duramin-40, Struers, Rotherham, UK) under a load of 200 gf (0.2 HV) and at a loading time of 10 s. Twelve indentations were made for the densest and thickest specimen, and the HV values were averaged after excluding the maximum and minimum values.

## 3. Results and Discussion

Regarding the choice of the gas mixture and the current, we were guided by the work of Kitamura et al. [6] and by our preliminary experiments. To determine the optimum condition based on this guidance, we used the four gas ratios of $Ar/N_2/H_2$ of 90/54/36, 81/81/18, 100/100/0, and 140/60/0 standard liters per minute (slm) with respective currents of 230, 180, 230, and 200 A. Under these conditions, the corresponding plasma powers were 112, 83, 67 and 59 kW, respectively. The suspension feeding rate was fixed at 45 standard cubic centimeters per minute (sccm). These and other processing parameters are shown in Table 1.

**Table 1.** Processing parameters used during coating of YOF containing $Y_2O_3$ or $Y_5O_4F_7$ by SPS.

| Parameters | Conditions | | | |
| --- | --- | --- | --- | --- |
| | (a) | (b) | (c) | (d) |
| Electric power, kW | 112 | 83 | 67 | 59 |
| $Ar/N_2/H_2$ flow rate (slm) | 90/54/36 | 81/81/18 | 100/100/0 | 140/60/0 |
| Total gas flow rate (slm) | 180 | 180 | 200 | 200 |
| Arc current (A) | 230 | 180 | 230 | 200 |
| Feeding rate (sccm) | 45 | | | |
| Atomizing gas flow rate (slm) | 15 | | | |
| Stand Off Distance (mm) | 50 | | | |
| Suspension concentration | 10 wt.% | | | |
| Solvent | Deionized water | | | |
| Transverse speed | 1000 mm/s | | | |
| Scan time (coating cycles) | 20 | | | |
| Substrate material | Al 6061 | | | |

The noticeable differences in the conditions of Table 1a through Table 1d are the plasma power, which is related to the arc current, hydrogen gas fraction and total gas flow rate. Chakravarthy et al. [27] reported that the higher the plasma power is, the higher the plasma temperature becomes, which increases the temperature and the velocity of the injected particles. Therefore, a high plasma power causes the particles to melt better and allows for molten particles to spread and flow better on the surfaces of the substrates, increasing the coating thickness, decreasing the porosity, and improving the interfacial bonding. The hydrogen fraction of the total gas is the main parameter affecting the plasma power. In our experiment, the hydrogen fractions of the total gas in Table 1a, b, c, d were 20 %, 10 %, 0 % and 0 %, respectively. The higher the hydrogen fraction is, the higher the plasma power becomes, as shown in Table 1. This is because $H_2$ has relatively high gas mass enthalpy and thermal conductivity values relative to those of other gases [28].

The total gas flow rate is related to the atomizing effect. The atomizing process fragments the liquid, making the droplet size smaller to allow for the particles to melt more easily. Atomizing is

greatly affected by the Weber number (We), which is defined as the ratio of the aerodynamic force of the plasma gas to the surface tension of the liquid [29,30].

$$We = \frac{\rho_g \times u_r^2 \times d_1}{\sigma_1}$$  (1)

Here, $\rho_g$ is the gas mass density ($kgm^{-3}$), $u_r$ is the relative velocity between the gas and liquid drop ($ms^{-1}$), $d_1$ is the diameter of the droplet or liquid jet (m), and $\sigma_1$ ($Nm^{-1}$) is the surface tension of the liquid.

Eq. (1) indicates that the Weber number is proportional to the gas mass density and the square of the relative velocity of the liquid and gas and the mean diameter of the droplet. It is also inversely proportional to the surface tension of the liquid. The larger the Weber number is, the greater the atomizing effect becomes. In Table 1, the surface tension of the liquid is identical in all four conditions and the droplet size might be similar among the four conditions because the total flow rates of the four conditions are similar. Therefore, the Weber number in Eq. (1) would be proportional to the plasma gas density and square of the plasma gas velocity.

The information of the plasma gas density and velocity is needed to estimate the Weber number. The mass densities of the plasma gases Ar, $N_2$ and $H_2$ are 1.661, 1.165, and 0.0899 kg/m³, respectively. In Table 1a and b, the gas mixture contains $H_2$ with a low gas mass density, and the total flow rate, directly related to the gas velocity, is 180 slm, which is less than 200 slm, which is the total flow rate in Table 1c and d. Therefore, the Weber numbers under the conditions with a total gas flow rate of 180 slm are less than those under the conditions with a total gas flow rate of 200 slm and the atomizing effects in the latter is greater than those in the former.

As mentioned above, the plasma powers under the conditions with a total gas flow rate of 180 slm were greater than those under the conditions with a total gas flow rate of 200 slm because hydrogen has higher gas mass enthalpy and higher thermal conductivity than other gases. In summary, the plasma powers in Table 1a and b are higher than those in Table 1c and d and the atomizing effects in Table 1a and b are weaker than those in Table 1c and d.

Although the Weber numbers predict that the process conditions of Table 1c and d produce denser and thicker coatings than those of Table 1a and b, the SEM images in Figures 2 and 3 show that the coatings which are deposited under the conditions in Table 1a and b are denser and thicker than those deposited under the conditions in Table 1c and d. These results show that the plasma power is more effective than the atomizing effect with regard to particle melting.

Figure 2 shows cross-section images of YOF coatings deposited by SPS at different plasma powers. The coating thicknesses were $58 \pm 3.4$, $25.8 \pm 2.1$, $5.6 \pm 0.6$, and $0.93 \pm 0.4$ μm at plasma powers of 112, 83, 67 and 59 kW, respectively. The coating thickness was determined from cross-sectional SEM images and the thicknesses at ten points except for the maximum and minimum values which were averaged. The porosities were $0.15\% \pm 0.01\%$, $0.25\% \pm 0.01\%$ and $5.50\% \pm 0.40\%$ at plasma powers of 112, 83 and 67 kW, respectively. The porosity was obtained from cross-sectional SEM images and the porosity values at five points were averaged. In this experiment, an Al substrate of 10 mm × 10 mm × 10 mm was used.

Figure 3 shows the higher magnification FESEM cross-sectional images of the sample shown in Figure 2.

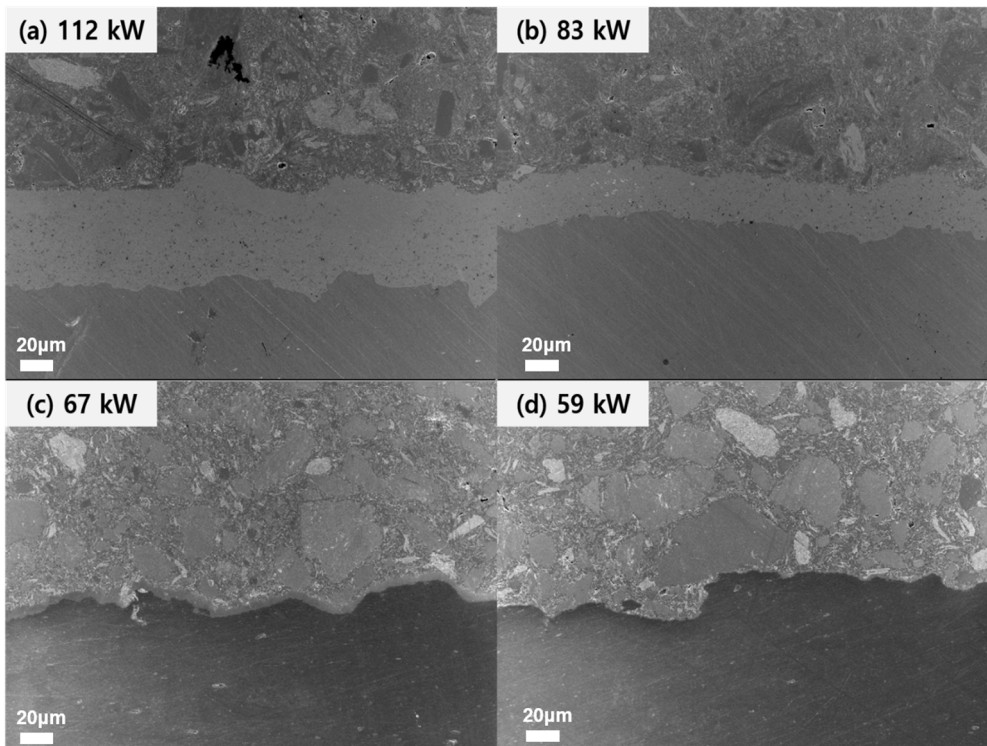

**Figure 2.** Cross-sectional FESEM images of YOF coatings deposited at plasma powers of (**a**) 112, (**b**) 83, (**c**) 67, and (**d**) 59 kW.

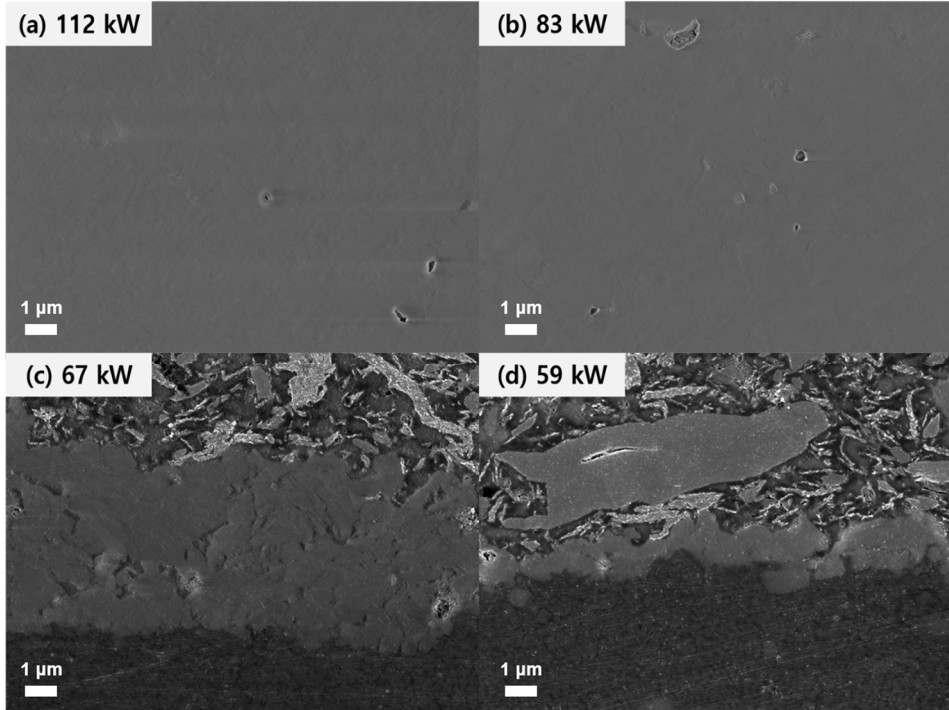

**Figure 3.** Higher magnification FESEM cross-sectional images of the sample shown in Figure 2.

Figure 4 shows the thickness and porosity of the YOF coating containing $Y_2O_3$ as a function of plasma power. The coating thickness increased and the porosity decreased with increasing plasma power.

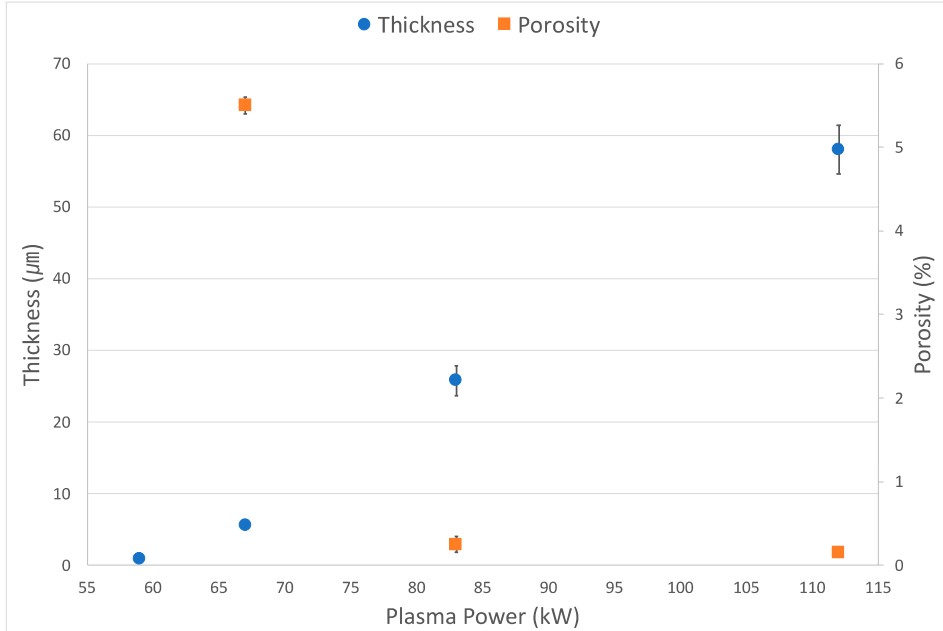

**Figure 4.** Thickness and porosity of the YOF coating containing $Y_2O_3$ as a function of plasma power.

The coating efficiency was estimated. The coating efficiency was regarded as the ratio of the mass of the deposited YOF coating containing $Y_2O_3$ to the mass of the supplied $Y_5O_4F_7$ particles in the suspension. The mass of the coating was calculated to be 6.75 g, considering a plasma gun scan area of 46 cm × 5 cm, coating thickness of 58 μm, and density of the trigonal YOF of 5.06 g/mL [31]. The mass of the supplied $Y_5O_4F_7$ was calculated to be 15.75 g, considering a suspension feeding rate of 45 mL/min, a feeding time of 3.5 min, a density of the suspension of the deionized water of 1 g/mL, and a $Y_5O_4F_7$ particle concentration of 10 wt%. From these values, the coating efficiency was estimated to be approximately 42.9 %.

When the coating is not thick enough, the measured hardness would be affected by the hardness of the substrate. For this reason, only the YOF coating which was deposited at 112 kW, as shown in Figure 2a, was chosen for the hardness measurement. The Vickers hardness of the YOF coating, which contains some $Y_2O_3$, was 550 ± 70 HV. This hardness is much higher than 290 ± 30 HV for the YOF coating reported recently by Lin et al. [16], who deposited a dense, low-porosity (0.5% ~1.0%) YOF coating by APS. On the other hand, Tsunoura et al. [4] reported that hot-pressed $Y_2O_3$ and YOF bulks had respective hardness values of 683.2 HV and 693.4 HV. Therefore, the hardness levels of YOF coatings that were deposited by SPS and APS referred to above are ~20 % and ~58 % lower than that of the aforementioned hot-pressed YOF bulks, respectively. Considering that the porosity of YOF coatings is relatively low at 0.14% ± 0.01% for the YOF coating in Figure 2a and 0.5%–1.0% for the YOF coating reported by Lin et al. [16], the hardness of the YOF coating being lower than that of the hot-pressed bulk does not appear to stem mainly from the porosity. One possible reason for the lower hardness would be the lack of the exact stoichiometry of the YOF coating, which is attributed to the rapid solidification of the molten YOF.

When the suspension is atomized and divided into fine droplets at the plasma torch, the solvent should be evaporated. Approximately 25% of plasma jet enthalpy is consumed in the plasma because the evaporation process is endothermic [29]. After evaporation, the $Y_5O_4F_7$ particles would be heated by the remaining thermal energy. If the remaining heat is not sufficient to melt the $Y_5O_4F_7$ particles, they would remain in a solid form and most of them would then bounce off from the substrate. In other words, non-melted $Y_5O_4F_7$ particles would not contribute to the deposition, but fully or partially melted $Y_5O_4F_7$ particles would do so. Therefore, as the plasma power is increased, the larger amount of $Y_5O_4F_7$ particles would be melted and contribute to the deposition. As a result, the deposition rate increases. This would explain why the coating thicknesses in Figures 2 and 3 increased with an increase in the plasma power.

There are several parameters affecting the porosity of the SPS coating such as droplet size, the size of particles in suspension, and incompletely melted particles [26]. Considering that the size of particles in suspension and the droplet size would be similar in all experimental conditions, the porosity of the SPS coating appears to mainly come from incompletely melted particles. Their amount would increase with decreasing plasma power. This would be why the porosity of the films in Figure 3 increased with decreasing plasma power. Therefore, the plasma power is critical and, in order to obtain thick and dense films by SPS, the plasma power should be high enough.

Figure 5a,b,c,d show the respective surface morphologies at plasma powers of 112, 83, 67 and 59 kW. The FESEM image presented in Figure 5a shows a microstructure consisting of smooth and rough areas. The smooth area would be splats which are formed from the spreading of molten droplets. The rough area would be formed either by fragments separated from the droplets when splashing on the growing surface or by incompletely melted particles. It should be noted that the total area of the smooth surface is much smaller in Figure 5d than in Figure 5a. This may stem from completely melted particles, which are far fewer in Figure 5d than in Figure 5a due to the low plasma power.

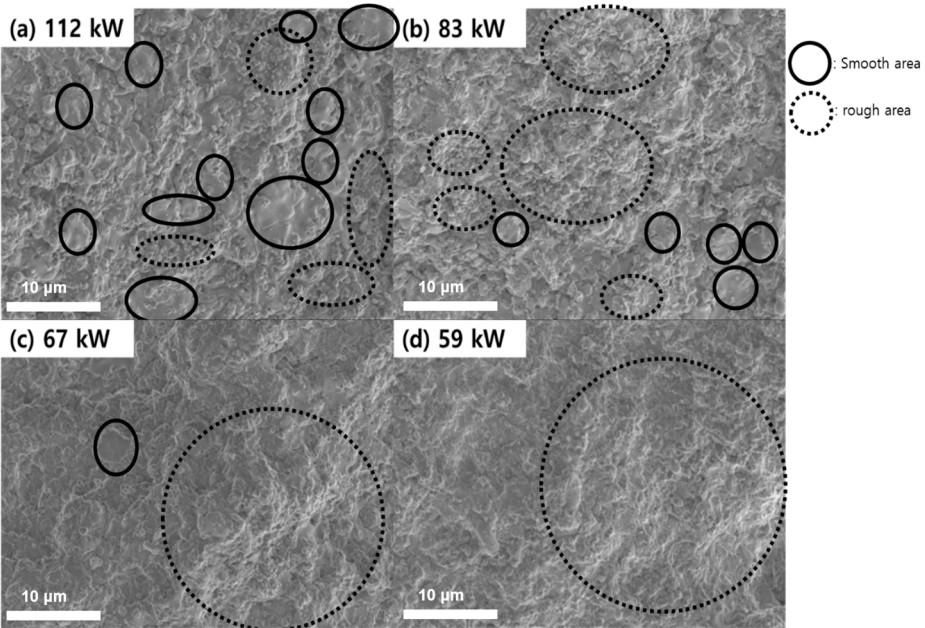

**Figure 5.** Surface FESEM images of YOF coatings deposited at plasma powers of (**a**) 112, (**b**) 83, (**c**) 67 and (**d**) 59 kW.

The crystalline structure of the YOF containing the $Y_2O_3$ or $Y_5O_4F_7$ coatings was investigated by X-ray diffraction (XRD), as shown in Figure 6. The major phase of trigonal YOF and minor phases of cubic $Y_2O_3$ and monoclinic $Y_2O_3$ were observed in the sample that was deposited at 112 kW (Figure 6a). On the other hand, Figure 6b shows that the peaks of the Al substrate mainly consist of minor peaks of orthorhombic $Y_5O_4F_7$ and trigonal YOF. Figure 6a shows that cubic and monoclinic $Y_2O_3$ formed at the plasma power of 112 kW. With regard to the formation of $Y_2O_3$, Park et al. [32] reported that YOF powder started to lose fluorine in the form of $YF_3$ at 900 °C according to a thermogravimetric analysis, which was performed to study the high-temperature volatilization of YOF. Based on this result, they suggested that YOF can be volatilized in the form of $YF_3$ in a plasma jet at ~5000 °C or higher. They also suggested that $Y_2O_3$ is formed by the volatilization of $YF_3$ from YOF.

Given that we used $Y_5O_4F_7$ particles, the $Y_2O_3$ in Figure 6a must have been formed from $Y_5O_4F_7$, $Y_5O_4F_7$ would not be directly transformed into $Y_2O_3$ but would be initially transformed into YOF, after which the YOF would be transformed into $Y_2O_3$. Regarding the formation of YOF from $Y_5O_4F_7$, Biqiu et al. [33] reported that, when $Y_5O_4F_7$ was heat-treated, it transformed into YOF. According to our experiment, however, if $Y_5O_4F_7$ was not sufficiently heated, it did not transform into YOF. For

example, when the plasma power was 59 kW, which is relatively low, our XRD data in Figure 6b showed that $Y_5O_4F_7$ had major peaks and YOF had only minor peaks. Moreover, the coating thickness was only $0.93 \pm 0.4$ μm, as shown in Figure 2d. Considering the low growth rate at the power of 59 kW in Figure 2d and the major peaks of $Y_5O_4F_7$ in Figure 6b, it appears that most of the $Y_5O_4F_7$ particles did not melt and that the unmolten $Y_5O_4F_7$ particles were not transformed into YOF, which would be attributed to the low volatilization rate of unmolten $Y_5O_4F_7$ particles. To summarize, $YF_3$ is volatilized from molten $Y_5O_4F_7$ particles, transforming $Y_5O_4F_7$ into YOF. Then, $YF_3$ is further volatilized from YOF, which would produce $Y_2O_3$ in Figure 6a. Although $Y_2O_3$ is also known to be a plasma-resistant material, it is less resistant to plasma than YOF because $Y_2O_3$ tends to react with the fluorine gas in the plasma and produce $YF_3$ particles, which act as contaminants. In this sense, the formation of $Y_2O_3$ is not desirable and it should be minimized, for which the volatilization rate of $YF_3$ should be controlled. Systematic efforts would be needed to determine the optimum processing condition in order to minimize the formation of $Y_2O_3$. On the other hand, Figure 6b shows that the peaks of the Al substrate mainly consist of a minor peak of orthorhombic $Y_5O_4F_7$ without any peak of trigonal YOF. This would be attributed to the fact that the YOF coating was scarcely coated at 59 kW, as shown in the FESEM images in Figures 2d and 3d. Because it is difficult to obtain quantitative information from the XRD data in Figure 6, we measured the compositions of Y, O, and F by EDS for the four coatings in Figure 2. These outcomes are shown in Table 2.

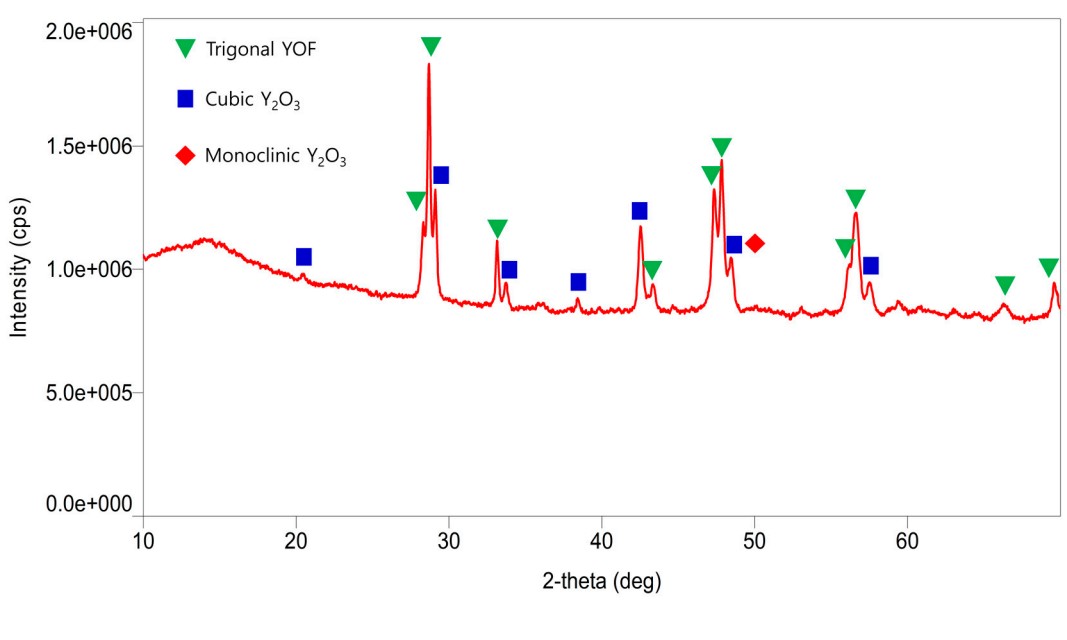

**(a)**

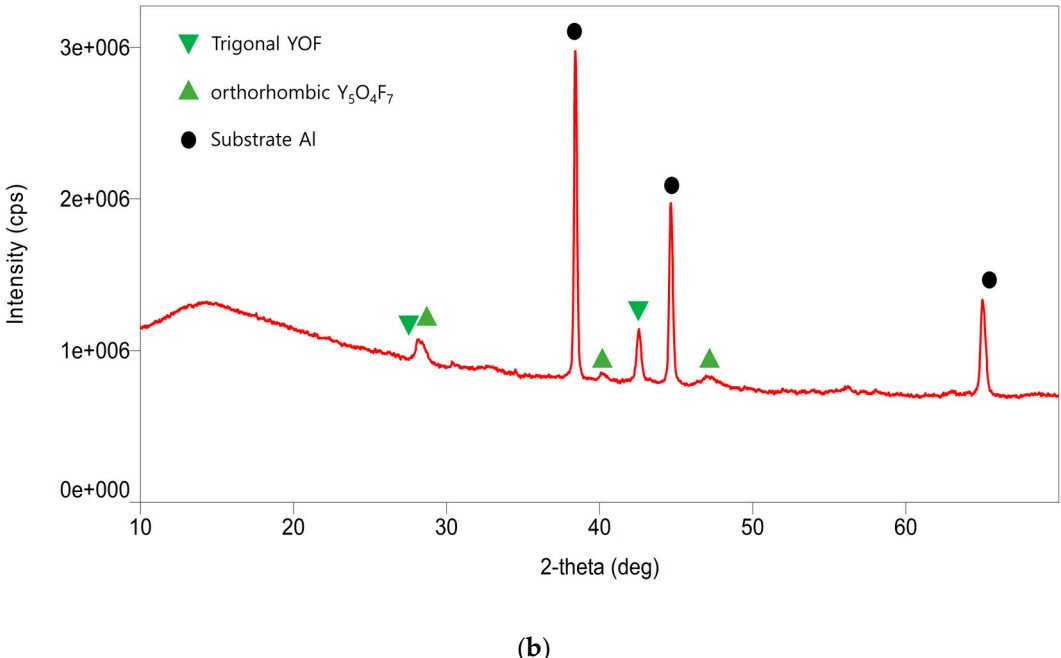

(**b**)

**Figure 6.** XRD patterns of YOF containing $Y_2O_3$ or $Y_5O_4F_7$ coatings deposited on Al substrates at plasma powers of (**a**) 112 kW and (**b**) 59 kW.

Table 2 shows that the atomic percentage of fluorine tends to decrease with an increase in the plasma power. This result supports our prediction that fluorine would be volatilized with an increase in the plasma temperature proportional to the plasma power. On the other hand, when considering that the atomic percentage of fluorine did not decrease above the plasma power of 83 kW, fluorine volatilization appears to be saturated above 83 kW.

**Table 2.** Compositions of YOF containing $Y_2O_3$ or $Y_5O_4F_7$ measured by Energy-Dispersive X-ray Spectroscopy (EDS).

| Atoms (at.%) | (a) 112 kW | (b) 83 kW | (c) 67 kW | (d) 59 kW |
|---|---|---|---|---|
| **Oxygen** | 46.39 | 44.57 | 39.45 | 28.46 |
| **Fluorine** | 24.09 | 22.16 | 30.15 | 43.42 |
| **Yttrium** | 29.52 | 33.27 | 30.40 | 28.12 |

As shown in Figure 7, EDS mapping of the coatings is qualitatively in agreement with the atomic percentage in Table 2. Fluorine, shown in green color, tends to decrease as the plasma power increases.

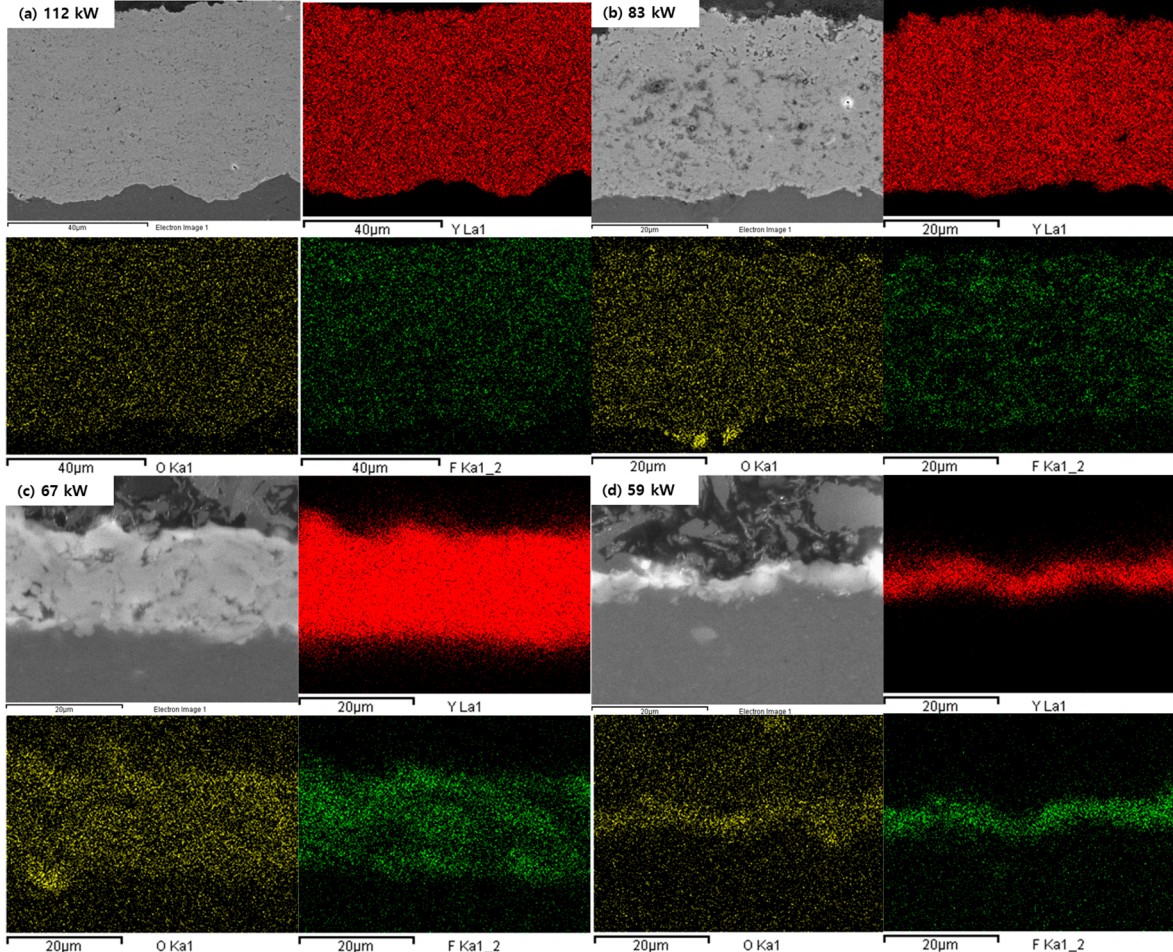

**Figure 7.** SEM-EDS elemental mappings of Y (red), O (yellow), and F (green) performed on the cross section of the YOF coatings containing $Y_2O_3$ or $Y_5O_4F_7$ deposited at (**a**) 112 KW, (**b**) 83 KW, (**c**) 67 KW, and (**d**) 59 KW.

In the SPS process, the size of the droplets, which are formed as a result of atomizing inside the torch, can affect the coating behavior [34]. If the droplet size is large, a large amount of solvent must be evaporated. In addition, the number of particles contained in the large droplet would also be large, which would then make full melting difficult. The particles contained in each droplet would become agglomerated as the solvent evaporates. Because the incomplete melting of agglomerates would contribute to the porosity and decrease the coating rate, smaller droplets would be favorable for a dense coating and a high coating rate.

One way to decrease the size of droplets would be to increase the flow rate of the atomizing gas. Lee et al. [34] reported that the droplet size was ~1 mm and a few hundred micrometers at the atomizing gas flow rates of 15 slm and 30 slm, respectively, because the atomizing gas flow rate affects the aerodynamic breakdown of the suspension. Because the maximum flow rate of an atomizing gas allowed in our equipment was 30 slm, we compared the coating behaviors using the two flow rates of 15 slm and 30 slm of the atomizing gas. In this experiment, an Al substrate as large as 50 mm × 50 mm × 10 mm was used to confirm a uniform coating on a large-area substrate. Other experimental conditions were identical to those in Figure 2a, except for the plasma power of 103 kW.

Figure 8a and b show FESEM images of YOF coatings deposited at rates of 15 slm and 30 slm of the atomizing gas, respectively. The deposition was successfully done on the entire area of the substrate. A great difference in the coating thicknesses between Figure 8a and b could not be found. However, the coating in Figures 8b and 9b, with porosity of 0.07% ± 0.03%, is denser than that in Figures 8a and 9a, for which the porosity is 0.22% ± 0.03%. The lower porosity of the coating in Figures 8b and 9b, as compared to that in Figures 8a and 9a, implies that a higher percentage of the agglomerate contained in the droplet underwent full melting at 30 slm, more than at 15 slm of the

atomizing gas. Because the incomplete melting of agglomerates is expected to decrease the coating rate, the coating in Figure 8b is expected to be thicker than that in Figure 8a. However, the coating thickness in Figure 8a was very close to the value of ~25 μm shown in Figure 8b. In order to explain this result, it is assumed that 15 slm of atomizing gas would produce partially melted agglomerates rather than unmelted agglomerates and that the partially melted agglomerates would contribute to the deposition.

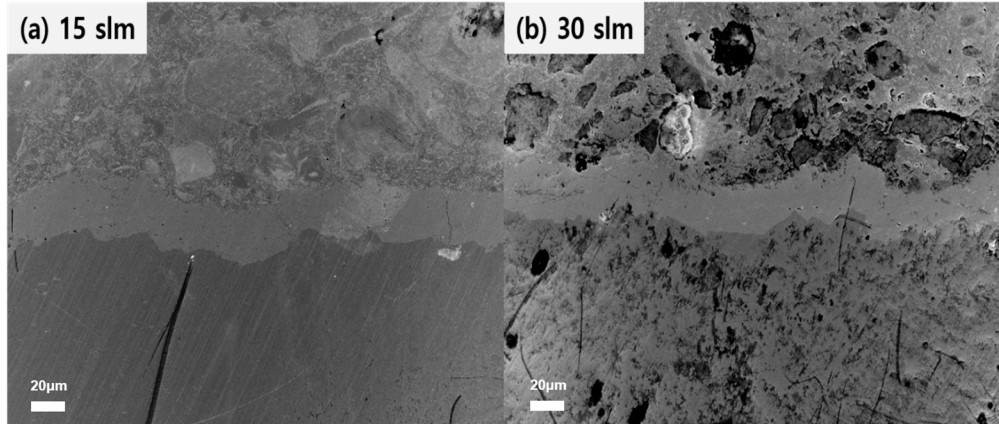

**Figure 8.** FESEM cross-sectional images of YOF coatings deposited on Al substrates by SPS at (**a**) 15 slm and (**b**) 30 slm of the atomizing carrier gas.

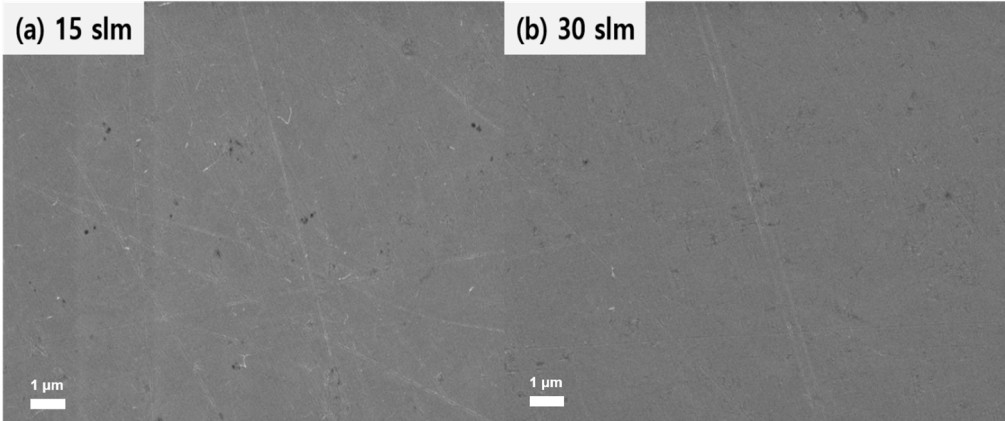

**Figure 9.** Higher magnification FESEM cross-sectional images of the sample shown in Figure 8.

## 4. Conclusion

Dense YOF-containing $Y_2O_3$ coatings could be successfully coated by SPS while using a high-output axial feeding method. As the plasma power was increased, the thickness of the coating increased and the porosity decreased. The 58 μm-thick film with the porosity of 0.15 % could be coated at the plasma power of 112 kW. The hardness of the coating deposited at 112 kW was 550 ± 70 HV, which is the highest value of the YOF coating by plasma spraying reported so far.

**Author Contributions:** Conceptualization, J.L. and S.L. and W.K. and N.-M.H.; Data curation, J.L.; Formal analysis, J.L. and S.L.; Funding acquisition, H.N.H and N.-M.H.; Investigation, J.L and S.L. and H.N.H and W.K and N.-M.H; Methodology, J.L and S.L.; Project administration, W.K. and N.-M.H.; Supervision, W.K and N.-M.H.; Writing - original draft, W.K.; Writing - review & editing, W.K. and N.-M.H. All authors have read and agreed to the published version of the manuscript.

**Funding:** This research was supported by the National Research Foundation of Korea (NRF) grant funded by the Korea government (MSIT) (No. 2020R1A5A6017701) and Samsung Electronics Co., Ltd.

**Conflicts of Interest:** The authors declare no conflict of interest.

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
