# Peer review of "Yttrium Oxyfluoride Coatings Deposited by Suspension Plasma Spraying Using Coaxial Feeding"

_coatings, doi:10.3390/coatings10050481_

Round 1

Reviewer 1 Report

In this paper Authors investigated the coating behavior of the yttrium oxyfluoride (YOF) coating deposited by suspension plasma spraying using a high-output coaxial feeding method. Authors investigated coating behavior with respect to the plasma power and the flow rate of the atomizing gas used.

The originality the concepts, the significance and the methods are good. The completeness and the organization of manuscript of the paper are good. The organization of the manuscript is good. In my opinion the technical treatment is plausible and free of technical errors.

Below I presented some remarks that came to my mind during reading:

  1. Line 23: It should be keywords.
  2. Each word in the name of the chapter/subchapter should be capitalized. Check in template.
  3. All aspects of the paper should be tailored to the Coatings template.
  4. In Introduction Authors should strongly emphasized the gap between previous studies and present studies.
  5. Values and units should be written separately.
  6. In Conclusions Authors must try to emphasize novelty, put some quantifications, and comment on the limitations. This is a very common way to write conclusions for academic journal. The conclusions should highlight the novelty and advance in understanding presented in the work.
  7. "Authors Contributions" and "Conflict of Interest" are missing.

Author Response

Replies to comments of reviewer # 1

Comment 1: Line 23: It should be keywords.

Answer: Thanks to the comment, we corrected the ‘key words’ to ‘keywords’.

Comment 2: Each word in the name of the chapter/subchapter should be capitalized. Check in template.

Answer: In response to the comment, we corrected the ‘Experimental procedure’, ‘Suspension plasma spray (SPS) system’, ‘Feedstock materials and specimens’ and ‘Analysis methods’ to the ‘Experimental Procedure’, ‘Suspension Plasma Spray (SPS) System’, ‘Feedstock Materials and Specimens’ and ‘Analysis Methods’, respectively.

Comment 3: All aspects of the paper should be tailored to the Coatings template.

Answer:  We revised the manuscript according to the Coatings template. We corrected the ‘Acknowledgments’ to ‘Funding’ and added the ‘Authors Contribution’ and ‘Conflict of Interest’.

Comment 4: In Introduction Authors should strongly emphasized the gap between previous studies and present studies.

Answer: In response to the comment, we revised the introduction as follows.

Old: Because coating YOF with the SPS method is currently in its early stages, the properties of the YOF coating are scarcely known. To the best of our knowledge, YOF coatings have not yet been implemented in the manufacturing process of semiconductors, although tests are underway regarding their adoption.

New: Because coating YOF with the SPS method is currently in its early stage, the properties of the YOF coating are scarcely known. To the best of our knowledge, YOF coatings have not yet been implemented in the manufacturing process of semiconductors, although tests are underway regarding their adoption.

In spite of the advantage of SPS, it has a critical problem of consuming lots of energy in evaporating the solution. Because of this energy consumption, the power should be high enough to melt the particles suspended in the solution. However, the plasma power cannot be increased too high because of the gouging of the electrode. To overcome this difficulty, we used the Axial III torch (Mettech), which has three cathodes and anodes with an axial feeding system. In this respect, the axial feeding has a much more advantage than the radial feeding commonly used in the SPS process due to the effective penetration of suspension into the core zone of the plasma jet. Besides, the high plasma power of the Axial III multi-electrodes has beneficial effects on particle melting and coating behavior.

Comment 5: Values and units should be written separately.

Answer: In response to the comment, we revised the manuscript. For example, we corrected the ‘0.15 ± 0.01%’, ‘67kW’ and ‘107’ in Abstract to ‘0.15 ± 0.01 ’, ‘67 ’ and ‘107 ’.

Comment 6: In Conclusions Authors must try to emphasize novelty, put some quantifications, and comment on the limitations. This is a very common way to write conclusions for academic journal. The conclusions should highlight the novelty and advance in understanding presented in the work.

Answer: Considering the comment, we revised the Conclusion as follows.

Old: Dense YOF-containing Y2O3 coatings could be successfully coated by SPS. As the plasma power was increased, the thickness of the coating increased and the porosity decreased.

New: Dense YOF-containing Y2O3 coatings could be successfully coated by SPS using a high-output axial feeding method. As the plasma power was increased, the thickness of the coating increased and the porosity decreased. The of 58 -thick film with the porosity of 0.15 % could be coated at the plasma power of 112 kW. The hardness of the coating deposited at 112 kW was 550 ± 70 HV, which is the highest value of the YOF coating by plasma spraying reported so far.    

Comment 7: "Authors Contributions" and "Conflict of Interest" are missing.

Answer: In response to the comment, we added the “Authors Contributions” and “Conflict of Interest” as follows.

Author Contributions
: Conceptualization, J.L. and S.L. and W.K. and N.-M.H.; Data curation, J.L.; Formal analysis, J.L. and S.L.; Funding acquisition, H.N.H and N.-M.H.; Investigation, J.L and S.L. and H.N.H and W.K and N.-M.H; Methodology, J.L and S.L.; Project administration, W.K. and N.-M.H.; Supervision, W.K and N.-M.H.; Writing - original draft, W.K.; Writing - review & editing, W.K. and N.-M.H. All authors have read and agreed to the published version of the manuscript.

Funding: This research was supported by the National Research Foundation of Korea (NRF) grant funded by the Korea government (MSIT) (No. 2020R1A5A6017701).

Conflicts of Interest: The authors declare no conflict of interest.

Reviewer 2 Report

The manuscript entitled “Yttrium oxyfluoride coatings deposited by suspension plasma spraying using coaxial feeding” reports the development of yttrium oxyfluoride coatings using plasma spraying technique. The objective of the study is good and the developed materials are adequately characterized and explained within the scope. Therefore, I recommend for the acceptance of the manuscript after the following moderate revision.

1. The plasma power vs thickness and porosity should be drawn as a graph for the better presentation

2. Provide the thermogravimetric data of the samples

3. Provide Vickers hardness data of the samples

4. Provide the EDS mapping of the samples

Author Response

Replies to comments of reviewer # 2

Comment 1: The plasma power vs thickness and porosity should be drawn as a graph for the better presentation                                                      

Answer: In response to the comment, we added the Figure 4 to Results and Discussion section and revised as follows.

Old: The porosity values were 0.15 ± 0.01 %, 0.25 ± 0.01 % and 5.5 ± 0.4 % at plasma powers of 112, 83 and 67 kW, respectively. The porosity was obtained from cross-sectional SEM images and the porosity values at five points were averaged. In this experiment, an Al substrate of 10 × 10 × 10 mm3 was used.

New: The porosity values were 0.15 ± 0.01 %, 0.25 ± 0.01 % and 5.5 ± 0.4 % at plasma powers of 112, 83 and 67 kW, respectively. The porosity was obtained from cross-sectional SEM images and the porosity values at five points were averaged. In this experiment, an Al substrate of 10 × 10 × 10 mm3 was used.                                                                                Fig. 4 shows thickness and porosity of the YOF containing Y2O3 coatings as a function of plasma power. As shown in Fig. 4, the coating thickness increased and the porosity decreased with an increase in the plasma power.

Comment 2: Provide the thermogravimetric data of the samples               

Answer: Thank you for the comment. But we could not carry out the thermogravimetric experiment of YOF powder in slurry, because we used all YOF slurry imported from abroad. If we order a new YOF slurry, it takes several months at least to arrive. Since we must finish revision process during 10 days, we could not do the experiment. However, we attached the figure from reference [31]( Park, S.-J.; Kim, H.; Lee, S.-M. Solid-State Synthesis of Yttirum Oxyfluoride Powders and Their Application to Suspension Plasma Spray Coating. Korean Journal of Materials Research 2017, 27, 710-715.), which would be similar with our case.

Comment 3: Provide Vickers hardness data of the samples                    Answer: We have worked with one engineering company. Because we provided our samples to the company for use in several property tests, unfortunately, we don’t have the samples can be used to measure hardness. But, as mentioned in Results and Discussion section, when the coating is not thick enough, the measured hardness would be affected by the hardness of the substrate. For this reason, only the YOF coating deposited at 112 kW, as shown in Fig. 2(a), was chosen for the hardness measurement. The hardness values of other samples deposited at lower plasma power would be smaller than that of the sample deposited at 112 KW due to the porosity.

Comment 4: Provide the EDS mapping of the samples

Answer: In response to the comment, we measured the EDS mapping of the samples and we added the Figure 7 (SEM-EDS elemental mapping performed on the cross section of the YOF containing Y2O3 or Y5O4F7 deposited at (a) 112 KW, (b) 83 KW, (c) 67 KW, (d) 59 KW) to the Results and Discussion section.

Old: Table 2 shows that the atomic percentage of fluorine tends to decrease with an increase in the plasma power. This result supports our prediction that fluorine would be volatilized with an increase in the plasma temperature proportional to the plasma power. On the other hand, considering that the atomic percentage of fluorine did not decrease above the plasma power of 83 kW, fluorine volatilization appears to be saturated above 83 kW.

New: Table 2 shows that the atomic percentage of fluorine tends to decrease with an increase in the plasma power. This result supports our prediction that fluorine would be volatilized with an increase in the plasma temperature proportional to the plasma power. On the other hand, considering that the atomic percentage of fluorine did not decrease above the plasma power of 83 kW, fluorine volatilization appears to be saturated above 83 kW.

Figure 7. SEM-EDS elemental mapping performed on the cross section of the YOF containing Y2O3 or Y5O4F7 deposited at (a) 112 KW, (b) 83 KW, (c) 67 KW, (d) 59 KW) to the Results and Discussion section.  <= see the manuscript

As shown in Fig. 7, EDS mapping data of the coatings shows the identical tendency to the atomic percentage in Table 2. Fluorine, shown in green color, tends to become sparse as plasma power increases.

Reviewer 3 Report

In this paper, the authors describe the use of suspension plasma spray technique to deposit YOF coatings. The authors analyze the morphology and composition by SEM, XRD and EDS. The authors demonstrate the effects of different plasma powers on the thickness, porosity, hardness and composition of the films deposited. This paper has some merit, but could be improved significantly with the following comments:

  1. The μm symbol is strange throughout the paper. Please use the correct symbol.
  2. On line 79, I think "sensible" should be "sensitive".
  3. Why were only two flow rates for the atomizing gas chosen? Is there a reason?
  4. How was the surface roughness measured? Throughout the paper, the authors do not provide roughness scans by AFM. It would be difficult to quantify roughness without AFM scans. SEM of rough samples may not provide an accurate method for measuring roughness.
  5. What does cooling at a distance of 100 cm mean?
  6. Symbols for × are also wrongly put as "x". Please use correct symbol.
  7. The proper reference for ImageJ should be given in the reference section as: Rueden, C.T.; Schindelin, J.; Hiner, M.C.; DeZonia, B.E.; Walter, A.E.; Arena, E.T.; Eliceiri, K.W. ImageJ2: ImageJ for the next generation of scientific image data. BMC Bioinformatics 2017, 18, 529.
  8. sccm and slm are actually standard cubic centimeter per minute and standard liter per minute. The word standard is important as it relates to pressure and temperature. Therefore it is strongly recommended that the authors add the word standard when defining the abbreviation.
  9. On line 191, would it not be easier to report the thickness either in mm or μm. Instead of 58 × 10-4 m, use either 5800 μm or 5.8 mm.
  10. SEM images in Figures 2, 3, 4, 6 and 7 do not have visible scale bars. It is imperative that the authors provide a visible scale bar.
  11. Analysis of rough and smooth in Figure 4 is not quantifiable. Please provide AFM scans of specific region to illustrate the roughness.
  12. Were XRD scans for 83 kW and 67 kW done? If not, it is recommended that these are added to improve the paper.
  13. It is immediately evident to me why a higher flow rate of the atomizing gas would decrease the droplet size. Can the authors add a small explanation?
  14. When reporting errors in values, it might be a better way to use the same number of decimal places. This applies throughout the text and also in the abstract.

Author Response

Replies to comments of reviewer # 3

Comment 1: The μm symbol is strange throughout the paper. Please use the correct symbol.

Answer: In response to the comment, we corrected ‘μm’ to ‘’.

Comment 2: On line 79, I think "sensible" should be "sensitive".

Answer: Thanks to the comment, we corrected “sensible” to “sensitive”.

Comment 3: Why were only two flow rates for the atomizing gas chosen? Is there a reason? Answer: Thank you for the comment. Although we’d like to test more than two conditions, due to the limitation in the setting range of the control unit software, we could test only two flow rates. The plasma equipment was the commercial product of National Research Center in Canada.

Comment 4: How was the surface roughness measured? Throughout the paper, the authors do not provide roughness scans by AFM. It would be difficult to quantify roughness without AFM scans. SEM of rough samples may not provide an accurate method for measuring roughness.

Answer: Thank you for the comment. We used MITUTOYO SJ-210 instrument to measure the substrate surface roughness. The purpose was to find out the optimum condition to increase the adhesion strength between substrate and coating. The operation principle of the instrument is similar to that of AFM. The measurement resolution is precise as three decimal places in micrometers. This measurement method is widely used to measure the surface roughness in industry. We revised our manuscript as follows.

Old: To improve the adhesion strength of the coatings, the surface of the Al substrate in each case was sandblasted to have a surface roughness average (Ra) value in the range of 2.3 ~ 2.8 μm by alumina particles (white fused alumina # 100, Dae Han Ceramics Co., Ltd.) less than 254  in size.

New: To improve the adhesion strength of the coatings, the surface of the Al substrate in each case was sandblasted to have a surface roughness average (Ra) value in the range of 2.3 ~ 2.8 μm by alumina particles (white fused alumina # 100, Dae Han Ceramics Co., Ltd.) less than 254  in size. The Ra of the Al substrate was measured by the surface roughness tester (MITUTOYO SJ-210).

Comment 5: What does cooling at a distance of 100 cm mean?

Answer: The substrate was cooled by the air gun at a distance of 100 cm, which was guided by references [6] to choose the distance, which used the same Axial â…¢ plasma torch as our experiment. We revised the manuscript as follows.

Old) In order to prevent the substrate damage due to the high-temperature plasma, the opposite side of the substrate was cooled by air at a distance of ~ 100 cm. The air flux coming from the air gun was wide enough to cool the 10 × 10 × 10 mm3 and 50 × 50 × 10 mm3 substrates.

New) In order to prevent the substrate damage due to the high-temperature plasma, the opposite side of the substrate was cooled by air at a distance of ~ 100 cm. We were guided by the reference [6] to choose the distance, which used the same Axial â…¢ plasma torch as our experiment. The air flux coming from the air gun was wide enough to cool the 10 × 10 × 10 mm3 and 50 × 50 × 10 mm3 substrates.

Comment 6: Symbols for × are also wrongly put as "x". Please use correct symbol.

Answer: In response to the comment, we revised “x” to “×” in the entire manuscript.

Comment 7: The proper reference for ImageJ should be given in the reference section as: Rueden, C.T.; Schindelin, J.; Hiner, M.C.; DeZonia, B.E.; Walter, A.E.; Arena, E.T.; Eliceiri, K.W. ImageJ2: ImageJ for the next generation of scientific image data. BMC Bioinformatics 201718, 529.

Answer: According to the comment, we inserted the reference for ImageJ.

Comment 8: sccm and slm are actually standard cubic centimeter per minute and standard liter per minute. The word standard is important as it relates to pressure and temperature. Therefore it is strongly recommended that the authors add the word standard when defining the abbreviation.

Answer: In response the comment, we revised the definition of slm and sccm as follows.

Old: Regarding the choice of the gas mixture and the current, we were guided by the work of Kitamura et al. [6] and by our preliminary experiments. To determine the optimum condition based on this guidance, we used the four gas ratios of Ar/N2/H2 of 90/54/36, 81/81/18, 100/100/0 and 140/60/0 liters per minute (slm) with respective currents of 230, 180, 230, and 200 A. Under these conditions, the corresponding plasma powers were 112, 83, 67 and 59 kW. The suspension feeding rate was fixed at 45 cubic centimeters per minute (sccm). These and other processing parameters are shown in Table 1.

New: Regarding the choice of the gas mixture and the current, we were guided by the work of Kitamura et al. [6] and by our preliminary experiments. To determine the optimum condition based on this guidance, we used the four gas ratios of Ar/N2/H2 of 90/54/36, 81/81/18, 100/100/0 and 140/60/0 standard liters per minute (slm) with respective currents of 230, 180, 230, and 200 A. Under these conditions, the corresponding plasma powers were 112, 83, 67 and 59 kW. The suspension feeding rate was fixed at 45 standard cubic centimeters per minute (sccm). These and other processing parameters are shown in Table 1.

Comment 9: On line 191, would it not be easier to report the thickness either in mm or μm. Instead of 58 × 10-4 m, use either 5800 μm or 5.8 mm.

Answer: In response to the comment, we changed the unit of thickness.

Old: The mass of the coating was calculated to be 6.75 g, considering a plasma gun scan area of 46 cm × 5 cm, coating thickness of 58 × 10-4 cm, and density of the trigonal YOF of 5.06 g/ml [30].

New: The mass of the coating was calculated to be 6.75 g, considering a plasma gun scan area of 46 cm × 5 cm, coating thickness of 58 , and density of the trigonal YOF of 5.06 g/ml [30].

Comment 10: SEM images in Figures 2, 3, 4, 6 and 7 do not have visible scale bars. It is imperative that the authors provide a visible scale bar.

Answer: In response to the comment, we revised the scale bars in Figures 2, 3, 4, 6 and 7.

Comment 11: Analysis of rough and smooth in Figure 4 is not quantifiable. Please provide AFM scans of specific region to illustrate the roughness.

Answer: As mentioned in the reply to comment 4, the roughness value of the Al substrate was measured by the surface roughness tester (MITUTOYO SJ-210). However, we didn’t measure the roughness of the coating surface. As mentioned earlier, we worked in collaboration with an engineering company. Since we sent our samples to the company for the test of several properties, the samples are not available now.

Since we must finish the revision of this paper within 10 days, we are sorry that we could not prepare samples in time.

Comment 12: Were XRD scans for 83 kW and 67 kW done? If not, it is recommended that these are added to improve the paper.

Answer: Thank you for the comment. But, as mentioned in the answer to the comment 11, we don’t have the samples can be used to measure XRD. Although we cannot provide XRD data for 83 kW and 67 kW, it is expected that the peaks of orthorhombic Y5O4F7 gradually disappear with increasing plasma power.

Comment 13: It is immediately evident to me why a higher flow rate of the atomizing gas would decrease the droplet size. Can the authors add a small explanation?

Answer: Considering the comment, we revised the manuscript as follows.

Old: One way to decrease the size of droplets would be to increase the flow rate of the atomizing gas. However, the maximum flow rate of an atomizing gas allowed in the equipment was 30 slm. Therefore, we compared the coating behaviors using the two flow rates of 15 slm and 30 slm of the atomizing gas.

New: One way to decrease the size of droplets would be to increase the flow rate of the atomizing gas. Lee et al. [34] reported that the droplet size was ~ 1 mm and a few hundred micrometers at the atomizing gas flow rates of 15 slm and 30 slm, respectively, because atomizing gas flow rate affects the aerodynamic breakdown of the suspension. Because the maximum flow rate of an atomizing gas allowed in our equipment was 30 slm, we compared the coating behaviors using the two flow rates of 15 slm and 30 slm of the atomizing gas.

Comment 14: When reporting errors in values, it might be a better way to use the same number of decimal places. This applies throughout the text and also in the abstract.

Answer: In response to the comment, we have unified the range of error. For thickness, the error range was unified to one decimal place, and for porosity, the error range was unified to two decimal places.

Round 2

Reviewer 3 Report

The authors have responded to my comments satisfactorily.